# Activation Mechanisms of the VPS34 Complexes

**DOI:** 10.3390/cells10113124

**Published:** 2021-11-11

**Authors:** Yohei Ohashi

**Affiliations:** MRC Laboratory of Molecular Biology, Protein and Nucleic Acid Chemistry Division, Francis Crick Avenue, Cambridge CB2 0QH, UK; yo@mrc-lmb.cam.ac.uk

**Keywords:** VPS34, PtdIns(3)P, VPS15, Beclin 1, ATG14L, UVRAG, lipids, membranes, autophagy, endocytic pathway

## Abstract

Phosphatidylinositol-3-phosphate (PtdIns(3)P) is essential for cell survival, and its intracellular synthesis is spatially and temporally regulated. It has major roles in two distinctive cellular pathways, namely, the autophagy and endocytic pathways. PtdIns(3)P is synthesized from phosphatidylinositol (PtdIns) by PIK3C3C/VPS34 in mammals or Vps34 in yeast. Pathway-specific VPS34/Vps34 activity is the consequence of the enzyme being incorporated into two mutually exclusive complexes: complex I for autophagy, composed of VPS34/Vps34–Vps15/Vps15-Beclin 1/Vps30-ATG14L/Atg14 (mammals/yeast), and complex II for endocytic pathways, in which ATG14L/Atg14 is replaced with UVRAG/Vps38 (mammals/yeast). Because of its involvement in autophagy, defects in which are closely associated with human diseases such as cancer and neurodegenerative diseases, developing highly selective drugs that target specific VPS34/Vps34 complexes is an essential goal in the autophagy field. Recent studies on the activation mechanisms of VPS34/Vps34 complexes have revealed that a variety of factors, including conformational changes, lipid physicochemical parameters, upstream regulators, and downstream effectors, greatly influence the activity of these complexes. This review summarizes and highlights each of these influences as well as clarifying key questions remaining in the field and outlining future perspectives.

## 1. Introduction

Phosphoinositides are phosphorylated forms of phosphatidylinositol (PtdIns). Phosphorylation takes place in one of the -OH groups of the inositol ring of PtdIns, which allows seven phosphoinositide species (three mono phosphates, three biphosphates, and one triphosphate) to occur. Although they exist at low abundance in the cell, their spatiotemporal regulation is vital for cell signalling, membrane traffic, and metabolic processes [1]. Among these lipids, phosphorylation at the 3-OH position is executed by phosphoinositide 3-kinases (PI3Ks). Based on their structures and substrate specificity, PI3Ks are divided into three classes, I, II, and III [2] (Figure 1). Class I PI3Ks have catalytic subunits p110α, p110β, p110γ, and p110δ and are encoded by the *PIK3CA*, *PIK3CB*, *PIK3CG* and *PIK3CD* genes, respectively. Class I enzymes synthesize PtsIns(3,4,5)P_3_ from PtdIns(4,5)P_2_ and are mostly active on the plasma membrane (PM) [2], although recent studies revealed that they also can be found on endosomal compartments [3,4]. By interacting with their regulatory subunits, they have important roles in signalling downstream of GPCRs and receptor tyrosine kinases [5]. Class II PI3Ks (PIK3C2α, PIK3C2β, and PIKC2γ, encoded by the *PIK3C2A*, *PIK3C2B*, and *PIK3C2C* genes, respectively) synthesize PtdIns(3,4)P_2_ from PtdIns(4)P [2,6,7], and have a minor contribution to PtdIns(3)P synthesis from PtdIns [2,8,9]. Class II PI3Ks can be found alone without a regulatory subunit in various compartments, including the plasma membrane, recycling endosomes [10], and late endosomes/lysosomes [6]. Class III has a sole member, VPS34/Vps34 (mammals/yeast), which synthesizes PtdIns(3)P from PtdIns. VPS34/Vps34 is the most ancestral PI3K, found in yeast and plants [11,12], whereas Class I and Class II PI3Ks can be found only in metazoans [2]. The *VPS34* gene was originally found in yeast (*S. cerevisiae*) by the Emr group as a gene in which mutations caused defects in the sorting of soluble vacuolar proteases but did not affect vacuolar morphology (“Class A” vpt (vacuole protein target) mutants) [13,14]. It is noteworthy that the *VPS15* gene was also found in the same screen and categorized in the same class [13]. Later, it was confirmed that these two genes and gene products functionally and physically interact [15,16]. The next breakthrough from the Ohsumi laboratory revealed that yeast Vps34 and Vps15 are part of two mutually exclusive complexes: complex I, composed of Vps34-Vps15-Vps30-Atg14, and complex II, composed of Vps34-Vps15-Vps30-Vps38 [17]. This one-subunit difference between the two complexes localizes them to different compartments, with complex I on the autophagosome, a double-membrane structure generated during starvation, and complex II on endosomal compartments. Later, orthologous mammalian complex I, composed of VPS34-VPS15-Beclin 1-ATG14L/Barkor [18,19,20,21], and complex II, composed of VPS34-VPS15-Beclin 1-UVRAG [18,20,22], were identified, indicating that not only the VPS34/Vps34 kinase subunit but whole heterotetrameric assemblies are conserved through evolution. As described below, each domain in the subunit of the complexes has a role in the assembly of the complex, interaction with lipids, or recruiting protein(s). These conserved heterotetrameric assemblies appear to be important not only for their localization, but also for efficient activation of the complexes. For example, the yeast Vps34–Vps15 heterodimer is less active than the yeast complexes I and II [23]. Also, human VPS34 alone is much less active than human complexes I and II [24]. These observations indicate that the heterotetrameric assembly is a minimal unit necessary for the full activation of VPS34/Vps34. The net activity of the complex is greatly influenced by conformational changes of the complex, upstream regulators, surrounding lipid environments, and feedback mechanisms from the downstream effectors. This review aims at summarizing the recent progress from studies of the activation mechanisms of VPS34/Vps34 complexes.

## 2. Basic Architectures of VPS34/Vps34 Complexes I and II

VPS34/Vps34, Vps15/Vps15, and Beclin 1/Vps30 (metazoans/yeast) are common subunits between complexes I and II, whereas ATG14L/Atg14 is complex I-specific and UVRAG/Vps38 is complex II-specific (Figure 2A). Because three of four subunits are shared between them, the structures of complex I and complex II have similar shapes, described as V-shaped or Y-shaped [23,25] (Figure 2B), with VPS34/Vps34-VPS15/Vps15 on one arm (the catalytic arm) and Beclin 1/Vps30-ATG14L/Atg14 or Beclin 1/Vps30-UVRAG/Vps38 on the other arm (the adaptor or regulatory arm) (Figure 2B). Recent progress in structural studies has revealed the role of each domain of each subunit in complexes I and II in the assembly and activation of the complexes. This is summarized in Figure 2A. For both complexes, the C2 domain of VPS34/Vps34 is located at the centre of the assemblies and serves as a structural hub that is unlikely to bind to membranes [23,25] (Figure 2B). The adaptor arms for both complexes are responsible for membrane binding and are essential for activation, although detailed features of the membrane-binding mechanisms appear to be different: complex I relies on the BATS domain in ATG14L [24,26,27,28], while complex II relies on the three motifs (aromatic fingers 1 and 2 and the hydrophobic loop) in Beclin 1/Vps30 [24,29,30,31]. Another important aspect is that the kinase domain of VPS34/Vps34 is autoinhibited by the kinase domain of VPS15/Vps15 [23,31,32]. It must be noted that except for its autophosphorylation in yeast [33], there has not been any VPS15/Vps15 substrate reported; therefore, whether VPS15/Vps15 is a more general protein kinase or not still remains elusive.

## 3. Activation Mechanisms of VPS34/Vps34 and Complexes I and II

### 3.1. Basic Architecture of the VPS34 Kinase Domain

Like other protein/lipid kinases, the kinase domain of VPS34/Vps34 is composed of N- and C-lobes, which are connected by a hinge region. ATP binds to a small pocket between these lobes (Figure 3A). The hinge is less conserved among PI3Ks, and the ATP-binding pocket of VPS34 is somewhat smaller than those of other PI3Ks. By using these VPS34-specific characteristics of the kinase domain, recent VPS34-specific inhibitors were designed to target the ATP-binding pocket [35,36,37,38,39,40] (Figure 3A). Also, the VPS34 kinase domain has common features for activity, the P-loop, activation loop, and catalytic loop (Figure 3B). The P-loop binds to phosphates of ATP, while the activation loop binds to the lipid substrate PtdIns [41]. In contrast to the other class I PI3Ks that have largely disordered activation loops in the structures of the wildtype enzymes, VPS34 has a completely ordered activation loop [41]. At the beginning of the activation loop, there is a highly conserved DFG motif, which is important for transferring the phosphate by binding metal ions. Point mutations in this motif abolish the kinase activity of the mouse VPS34 [42]. The catalytic loop of VPS34 has a conserved DHR motif whereby the gamma phosphate of ATP is transferred to the substrate PtdIns.

### 3.2. Activation Mechanisms of VPS34/Vps34 Complexes I and II

Complex II is recruited to early endosomes and activated there by an early endosome-residing small GTPase, Rab5 [31,43,44,45] (Figure 3B), which results in PtdIns(3)P enrichment in early endosomes [46,47]. While this recruitment and activation had been long known in the cell, the underlying mechanism was finally revealed recently using electron cryotomography (cryo-ET), hydrogen–deuterium exchange mass spectrometry (HDX-MS), and unnatural amino acid (UAA)-mediated cross-linking [31]. Without Rab5, complex II lies in an inactive state in which the VPS34 kinase domain is autoinhibited by the VPS15 kinase domain. Then, GTP-bound Rab5a recruits complex II to membranes mainly by binding to the C2 helical hairpin (C2HH) insertion in the C2 domain of VPS34. This Rab5a binding has a dual role: first, to recruit complex II to membranes, and second, to release the inhibition of the VPS15-mediated VPS34 kinase domain (Figure 3B middle). Membrane-bound complex II is still flexible enough to be able to transiently move the catalytic arm up and down to synthesize PtdIns(3)P from PtdIns (Figure 3B right). In contrast to human complex I (see below), this Rab5a binding does not appear to cause a dislodging effect of the VPS34 kinase domain, indicating that the activation mechanism can be different depending on the complexes or their binding proteins [48]. The same study also found a new interaction between complex I and Rab1a and showed that membrane-immobilized Rab1a greatly increased complex I activity [31] (Figure 3C, left). Interestingly, GTP-bound Rab1a also binds to the VPS34 C2HH, the same region that complex II’s VPS34 uses to interact with Rab5a. Nonetheless, Rab1a specifically activates complex I [31]. The detailed activation mechanism of complex I by Rab1a remains to be seen in the future. It must be noted that in Tremel et al. [31], the complex I–Rab1a interaction was detected and examined in rich-medium conditions. In the cell, there is evidence for the involvement of Rab1/Ypt1 in autophagy during starvation. In yeast, Ypt1 binds to Atg11 [49], Atg1 [50], and the autophagy-specific TRAPPIII complex, which is a GEF for Ypt1 [50,51]. Human Rab1 also binds to ULK1 [50]. Also, in both yeast and mammals, Rab1/Ypt1 was immunoisolated from ATG9/Atg9 vesicles [52,53]. The localization of yeast Ypt1 to the PAS was dependent on Atg9 and TRAPIII [52], and the association of Rab1 with ATG9 vesicles was independent of ATG14L [53]. While all of these studies showed the association of Rab1/Ypt1 with either the ULK1/Atg1 complex or ATG9/Atg9 vesicles, how Rab1/Ypt1 regulates complex I during autophagy/starvation remains to be seen. For human complex I, a negative stain EM analysis showed that 13% of particles lacked density for the HELical-CATalytic region (HELCAT) of VPS34 [32,41]. The authors hypothesized that this highly mobile HELCAT (“dislodging”) could be the key for the activity of the complex. However, the same group later reported different structures of VPS34 complex I at higher resolutions using cryo-EM. Only the VPS34 kinase domain was mobile in apo complex I and in complex I in the presence of the complex I-specific binding subunit NRBF2 [48] (Figure 3C right; for NRBF2, see below). No dislodging effect of the HELCAT was found in this study, and the authors speculated that dislodging is an intermediate state [48]. Using the autoinhibition information, Steinfeld et al. designed active mutants for the yeast Vps34–Vps15 heterodimer by introducing point mutations at an interface between the Vps34 HELCAT and Vps15 kinase domain [54]. Unlike in human complex I, the kinase domains of Vps34 and Vps15 are tightly packed in the yeast Vps34–Vps15 heterodimer [55]. Also, the yeast Vps34–Vps15 heterodimer is not active on flat membranes and much less active on small (100 nm) vesicles than the yeast complexes I and II [23]. This indicates that the yeast Vps34–Vps15 might not be active because of the tight packing between the kinase domains of Vps34 and Vps15, which might be somehow relaxed by having full assemblies in heterotetrameric complexes.

## 4. Controlling the Intracellular PtdIns(3)P Concentration

In yeast, *vps34∆* and *vps15∆* mutants showed temperature sensitivity in the SEY6210 strain [14,15]. Knockout mice lacking the *PIK3C3* (encoding VPS34), *PIK3R4* (encoding VPS15), and *BECN1* (encoding Beclin 1) genes were all embryonically lethal [56,57,58,59], indicating that PtdIns(3)P is essential for cell survival and intracellular homeostasis of yeast and mammals.

PtdIns(3)P distribution on the autophagosomal membranes is different between yeast and mammals. In yeast, PtdIns(3)P was more abundant on the luminal side than the cytoplasmic side, whereas in mammals, PtdIns(3)P was predominantly detected on the cytoplasmic side of the autophagosome [60]. Because the asymmetrical distribution of PtdIns(3)P in yeast can be partially recovered by PtdIns(3)P phosphatase mutations [60], it appears that in yeast, the PtdIns(3)P on the cytoplasmic side of the autophagosome is quickly metabolized into PtdIns. Consistently with this, autophagy is defective in *ymr1∆* yeast cells, which lacks the myotubularin (MTMR)-type phosphatase gene *YMR1* [61]. In sharp contrast, in mammals, PtdIns(3)P phosphatases Jumpy (MTMR14), MTMR3, and the MTMR8–MTMR9 complex (see below) negatively regulate autophagy, indicating a positive correlation between the PtdIns(3)P amount on the autophagosome and autophagy activity [62,63,64]. This difference in the distribution of PtdIns(3)P may reflect the necessity of PtdIns(3)P for the downstream pathway during autophagy, i.e., in mammals, there might be more downstream PtdIns(3)P-binding effectors than in yeast. Alternatively, in mammals, the PtdIns(3)P generated on the autophagosome might be more readily used as a precursor for other phosphoinositides (also see Section 8).

Regulation of intracellular PtdIns(3)P concentration appears to be important for human health. Myotubularins comprise 15 members, MTM1 and MTMRs 1–14. Among them, 9 members are catalytically active and dephosphorylate PtdIns(3)P and PtdIns(3,5)P_2_ to convert them to PtdIns and PtdIns(5)P, respectively. Some myotubularins are essential for the maintenance of muscles [65,66], and mutations in them are associated with neuromuscular diseases. Mutations in the *MTM1* gene caused a severe form of X-linked congenital myopathy [67]. Also, mutations in the *MTM2* gene were found in Charcot–Marie–Tooth disease [68]. PtdIns(3)P levels were increased in the *mtm1* null murine muscle [69] and in HeLa cells treated with siRNA to deplete *MTM1* [70]. Also, muscles from *Mtm1*-null mice showed increased autophagic activity [71]. The *Mtm1* KO mouse phenotype was rescued by muscle-specific ablation of *Pik3c2b*, the class II PI3K, which also contributes to PtdIns(3)P production (see Section 1 and Figure 1) [9,72]. The same study also showed that the *mtm1* phenotype in zebrafish was rescued by PtdIns(3)P inhibitor treatments [72]. In addition, defects in cells from cross-linked centronuclear myopathy patients could be partially rescued by treatment with a PtdIns(3)P inhibitor [70]. These observations indicate that excessive PtdIns(3)P, in addition to insufficient PtdIns(3)P, are both toxic to the cell. Interestingly, conditional knockout mice lacking *Vps15* in skeletal muscles also showed a severe myopathy [57]. Intracellular PtdIns(3)P concentration appears to be tightly regulated to not be too high or too low.

## 5. Regulation of VPS34/Vps34 Complex Activity by Phosphorylation from Upstream Regulators, the ULK1 Complex and mTORC1

Both in yeast and metazoans, complex I activity is regulated by various upstream regulators. The ULK1/Atg1 complex and the mTOR/TOR complex (mammals/yeast) are two of the best-characterized regulators. ULK1/Atg1 and mTOR/TOR are serine/threonine kinases that phosphorylate autophagy-related (ATG/Atg) proteins including the complex I subunits to activate or inhibit the initiation step of autophagy [73,74] (Figure 4A). Although there are many more upstream regulators [35], including AMPK1 [75,76] and Akt [77], this section focuses on these two regulators.

### 5.1. Complex I Regulation by Phosphorylation from the ULK1/Atg1 Complex

Macroautophagy (hereafter autophagy) is triggered by various starvation conditions. Autophagy comprises several steps, the first of which is initiation, followed by nucleation, during which PtdIns(3)P made by complex I, potentially on the ER, recruits downstream effectors such as DFCP1 and WIPIs (see Section 8). This nucleates isolation membranes, or phagophores. Isolation membranes are then expanded and closed to become an autophagosome. The autophagosome is fused with lysosomes to become autolysosomes, after which the internal cargos or substrates are degraded to regenerate amino acids, lipids, and nucleic acids.

The ULK1/Atg1 complex is essential for the initiation of all types of canonical autophagy. The ULK1/Atg1 is composed of ULK1-FIP200-ATG101-ATG13 for mammals and Atg1-Atg13-Atg17-Atg29-Atg31 for yeast [78]. It is genetically located at the most upstream point of a hierarchy of ATG/Atg proteins that gives rise to the preautophagosomal structure or the phagophore assembly site (PAS) [79,80]. In the absence of the ULK1/Atg1 complex, complex I cannot be recruited to the PAS [80,81]. Although this complex I recruitment mechanism is still not fully understood, ULK1 phosphorylation on complex I subunits during autophagy has been associated to this process. Remarkably, all of the complex I and II subunits have been reported to be phosphorylated by ULK1 [73,82,83,84], indicating the importance of ULK1 on the regulation of complex I activity. Russell et al. found that the S15 residue of Beclin1 (note that the original paper reported this as S14) was phosphorylated by ULK1, and that this phosphorylation was critical for autophagy induction by amino acid starvation [82]. Also, the authors showed that S15 phosphorylation was promoted by either ATG14L or UVRAG [82]. The S30 residue on Beclin1 was also reported to be phosphorylated by ULK1 during amino acid starvation and hypoxia [83]. While this phosphorylation was promoted by ATG14L, it was not by UVRAG [83]. Yeast Vps30 was found to be phosphorylated by Atg1 [85,86], specifically at S85 [85], although its role has not been characterized. On the complex I-specific ATG14L subunit, S29 phosphorylation by the ULK1 complex was increased by starvation and mTOR inactivation [84]. This phosphorylation was important for the synthesis of PtdIns(3)P and initiation of autophagy, but not for the maturation step [84]. On VPS34, the S249 residue was shown to be phosphorylated by ULK1, although neither phosphomimetic nor nonphosphorylatable mutants significantly affected autophagy [87]. On the complex I-specific fifth subunit NRBF2/Atg38 (see Section 9), multiple serine residues (S96, S103, S109, and S133) on Atg38 were found to be phosphorylated by Atg1, although the role(s) of the phosphorylation on these residues have not been characterized [85]. Also, NRBF2 was phosphorylated by ULK1, although the exact site of this phosphorylation is not clear [88]. Lastly, a recent unbiased phosphoproteomics study in search of ULK1 and ULK2 substrates using SILAC and tandem mass tag (TMT) found six phosphorylation sites on VPS15 (S813, S861, S865, S879, S1039, and S1289) [89]. Mutations of these residues affected autophagy and PI3K activities in vivo and in vitro, with S861 playing the main role among the six phosphorylation sites [89]. The same study also found that UVRAG was phosphorylated by ULK1, though the exact amino acid residue and its role remain to be seen [89].

Understandably, the phosphorylation of each subunit/residue was characterized individually and separately. However, the phosphorylation by ULK1/Atg1 on all subunits should take place at the same time in the cell. This could be a reason why the single S249 mutation on VPS34 did not show a significant effect. With the availability of information on the ULK1 phosphorylation of all individual complex I and II subunits, it will be interesting to examine the effect of the sum or synergy of the phosphorylations on the whole complexes in vitro. One technical difficulty for in vitro experiments is that overexpression of full-length ULK1 causes toxicity to the cell [90,91,92]. Therefore, the insect cell-expressed soluble ULK1 kinase domain, which is also toxic when expressed in bacteria [93], has been used for in vitro ULK1 kinase assays. This may lack the substrate recognition domain/motif(s) of ULK1 and may result in less specific substrate recognition, as regulatory subunits of the ULK1 complex (FIP200 or ATG13) have roles in specifying substrates. For instance, the lack of ATG13 or FIP200 abolished the S30 phosphorylation of Beclin1 [83] and S29 phosphorylation of ATG14L [84]. Immunoprecipitated endogenous ULK1 complex has also been used for the ULK1 kinase assays, as it can eliminate the problem described above. However, the protein identities of the immunoprecipitants are less clear, and effects from contaminated proteins cannot be excluded. It appears that coexpression of ULK1-ATG101-ATG13 in mammalian suspension cells can overcome this problem [94]. Also, full-length yeast Atg1 alone can be expressed in insect cells and reconstituted with Atg13 and the Atg17 complex [95,96,97]. These may be useful solutions for designing more precise experiments.

The potential effect of the ULK1 phosphorylation on the complex I subunits might be conformational changes and/or an increase in local charge, which may lead to direct interaction with the ULK1 complex or other proteins or complexes. For example, immunoprecipitation studies detected an interaction between mammalian ATG13 and ATG14L [84,98], whereas in yeast this interaction was not detected [99].

### 5.2. mTORC1–VPS34 and VPS34–mTORC1 Pathways

Autophagy induction by rapamycin was originally shown for yeast [100]. Later, this mechanism was found to be conserved through evolution [101,102,103]. Rapamycin inactivates mechanistic/mammalian target of rapamycin (mTOR), or TOR in yeast, which mimics starvation conditions to trigger autophagy. mTOR serves as a hub in the nutrient signalling network. It can be found in two mutually exclusive complexes, mTORC1 and mTORC2, which are composed of mTOR-Raptor-mLst8 and mTOR-Protor-Rictor-mSin1-mLst8, respectively. mTORC1 is acutely inactivated by rapamycin [104,105], whereas the effect of rapamycin on mTORC2 is dependent on cell line, tissue, and duration of treatment [106,107]. mTOR is activated by amino acids, growth factors, and insulin to phosphorylate various substrates [108]. In particular, mTORC1 phosphorylates autophagy-related proteins to inactivate them in rich-medium conditions [74,109]. The formation of the yeast Atg1 complex is prevented by TORC1 in rich medium by hyperphosphorylation of Atg13 [110,111], which impairs PAS formation. On the other hand, the formation of the mammalian ULK1 complex is not affected by nutrient condition [101], although ATG13 is similarly phosphorylated by mTOR in nutrient-rich conditions as the yeast Atg13 by TOR [76,101]. Whereas the ULK1 complex negatively regulates mTORC1 activity [112]. Thus, two upstream autophagy regulators, the ULK1/Atg1 complex and the mTOR/TOR complex 1, mutually regulate each other (Figure 4A).

Interestingly, among the complex I and II subunits, only the complex-specific subunits (ATG14L, NRBF2, and UVRAG) have been found to be phosphorylated by mTORC1. Multiple serine/threonine residues in ATG14L were found to be phosphorylated by mTOR, and mutating these residues into alanines increased PI3K and autophagy activity [113]. The mTOR phosphorylation of complex I-specific NRBF2 on the S113 and S120 residues leads NRBF2 to bind to VPS34–VPS15 subunits and inactivate the complex [88]. In sharp contrast, an in vitro study showed that bacterially purified, PTM-free NRBF2 bound to VPS15 to activate complex I [114]. Reconciling these opposite results remains to be done in the future. On the other hand, dephosphorylation of these residues upon starvation rearranges the NRBF2–complex I interaction into an NRBF2–Beclin1–ATG14L interaction in the complex, in turn activating the complex [88]. Three serine residues in complex II-specific UVRAG are known to be phosphorylated by mTOR: S498, S550, and S571. S498 phosphorylation increases the association with Rubicon, the fifth subunit of complex II (see Section 9). This leads to a decrease in VPS34 activity and, consequently, inhibition of endosome and autophagosome maturation steps [115]. On the other hand, S550 and S571 phosphorylation increase complex II activity [116]. Moreover, this activation is required for the scission step of lysosomal tubulation during autophagosome–lysosome reformation (ALR) [116,117] (see Figure 4B for more details). All of these serine residues are located in the metazoan-specific C-terminal domain of UVRAG (CTD, Figure 2 and Figure 4B). Because structural information is not available for this region, the mechanisms of these inhibition/activation events remain to be seen. Similarly as for ULK1 phosphorylation, the sum or synergy of the mTOR phosphorylation on all complex I and II subunits will reveal more detailed mechanisms for the activation and conformational changes of the complexes.

VPS34 is known to increase mTOR activity. This is mainly based on the findings that amino acids or insulin increased VPS34 activity, which was followed by an increase in mTOR activity [118,119,120], and the findings that mTOR was activated by phosphatidic acid (PA) [121,122,123], which is synthesised by phospholipase D (PLD) from phosphatidylcholine (PC, Figure 4B). PLDs possess a PX domain that binds to PtdIns(3)P (for more details on the PX domain, see Section 8.3). PtdIns(3)P increased PLD1 activity via its PX domain [124]. In addition, mice with defects in the *Mtm1* gene, the product of which removes the phosphate from PtdIns(3)P, showed hyperactivation of mTOR [125] (see Section 4). An NMR study showed that PA bound to the FRB domain of mTOR [126]. Based on the structures of mTORC1 in complexes with RagA/C heterodimers, and with the small G-protein RHEB, it was proposed that the FRB domain is located distantly from the membrane, where PA is situated [127]. One possibility is that PA may also bind to some other region(s) of mTORC1. Alternatively, mTORC1 may undergo conformational changes to bind to PA. Also, Raptor binds to PLD2 via the TOS-like motif on PLD2 [128]. This amino acid-stimulated VPS34–mTOR activation was found to be mediated by leucyl-tRNA synthase (LRS) in the presence of leucine [129]. LRS binds to VPS34 in vivo, which facilitates VPS34 activation [129] (Figure 4B).

Another proposal about mTOR activation by VPS34 is that the PtdIns(3)P-binding FYCO1 translocates the mTORC1-carrying lysosomes to the cell periphery to facilitate mTOR activation [130] (see Figure 4C for more details).

In contrast, when human skeletal muscle cells were serum starved, then subsequently serum recovered, VPS34 inhibition did not prevent mTOR activity [131]. Also, in steady-state mouse embryonic fibroblasts (MEFs), Vps34 deficiency did not affect mTOR activity, but Vps34 was essential for acute mTOR activation [132]. This indicates that mTOR activation by VPS34/PtdIns(3)P might be dependent on cell type or growth condition.

## 6. Genetic and Cell Biological Hierarchy of the Recruitment of Complex I at the Initiation Step of Autophagy

As described above, the ULK1/Atg1 complex is an essential upstream regulator for complex I and for the initiation of autophagy. An ultrastructural analysis showed that isolation membranes could not be formed in FIP200 KO or ATG9 KO MEFs, and that in HeLa cells treated with a PI3K inhibitor, wortmannin, the selective autophagy substrate p62 was aggregated, and unknown small vesicles were clustered at the autophagosome formation site [81]. This indicates that the ULK1 complex I, ATG9, and PtdIns(3)P (complex I) are essential for the initiation of autophagy. Several hierarchical analyses of these three components have revealed that the ULK1 complex and ATG9 are essential for the localization of ATG14L (and hence complex I) [80,81]. The ULK1 complex targets some ER-related membranes other than ATG9 vesicles independently of PtdIns(3)P, since the ULK1 complex subunits can associate with membranes in the absence of ATG9 [53]. In addition, ULK1 can target p62 in ATG9 KO MEFs [81]. Membrane associations of the ULK1 complex subunits and ATG9 are independent of ATG14L [53]. FIP200, an ULK1 complex subunit, tethers the other ULK1 component to membranes independently of ATG9 and PtdIns(3)P [53,81], and ATG9 association with membranes and p62 is also independent of the ULK1 complex and PtdIns(3)P [81]. These observations suggest that the ULK1 complex and ATG9 are upstream of complex I, and that the ULK1 complex and ATG9 may contribute in parallel to the initiation step. Similarly, in yeast, Atg11 and Atg17, which are the Atg1 complex subunits with scaffolding roles, are at the most upstream end of the hierarchy [79]. Although FIP200 was initially proposed as the Atg17 orthologue [92], structurally, Atg11 is closer to FIP200 [133]. Also, Atg9 is at the upstream of yeast complex I [79]. During Parkin-mediated mitophagy, the ULK1 complex and ATG9 target depolarized mitochondria independently, and a similar hierarchical recruitment of the downstream ATG proteins was reported [134]. Emerging evidence has suggested that the ULK1 complex targets two distinct membranes: ER membranes and ATG9 vesicles. PtdIns(3)P is required for the translocation of the ULK1 complex to ATG9-positive membranes [53]. Also, upon treatment by wortmannin, the lifespan and size of ATG13-positive puncta become shorter and smaller, respectively [135]. These results indicate that although PtdIns(3)P/complex I is dispensable for the initial localization of the ULK1 complex and ATG9, it may become important for maintaining the ULK1 complex on membranes to create a positive feedback loop and for proceeding to downstream steps of autophagy. It remains to be seen how PtdIns(3)P or complex I can contribute to the maintenance mechanism of the ULK1 complex.

## 7. Lipid Environments That Influence the VPS34/Vps34 Complex Activities

### 7.1. Physicochemical Parameters That Affect the VPS34/Vps34 Activities

Because phospholipids are major constituents for biological membranes, they are the focus here. A lipid molecule is composed of a polar head, a backbone (glycerol or sphingosine), and fatty acids or acyl chains (Figure 5A). The polar head is composed of an alcohol moiety and a phosphate, and the alcohol substituent defines the lipid species, such as phosphatidylcholine (PC), phosphatidylethanolamine (PE), phosphatidylserine (PS), and PtdIns. Polar heads face towards the aqueous side in a lipid bilayer and thus have a role in binding to lipid-specific binding proteins. For instance, the polar heads of PS and phosphoinositides are negatively charged. Therefore, they can attract various positively charged proteins or protein patches (see below). Despite the fact that lipids are constantly trafficked and shuffled among membrane compartments, each organelle has a relatively unique lipid composition represented by polar heads [136,137]. Also, acyl chain properties are tissue- and organelle-specific and greatly influence protein–membrane interactions [138,139,140]. Collectively, there are three important physicochemical parameters of a membrane that can affect membrane–protein interaction and lipid kinase/phosphatase activities: lipid packing of acyl chains, membrane curvature, and electrostatics of polar heads as represented by PS and phosphoinositides [140] (Figure 5B), which are explained below in greater detail.

### 7.2. Lipid Packing of Acyl Chains

The length and saturation status of acyl chains are key determinants to characterize biological membranes [141,142] (Figure 4B). Double bonds in acyl chains introduce kinks in the chains (Figure 5A), causing lipid-packing defects (Figure 5B). Packing defects provide more space in the lipid bilayer, which makes the membrane more flexible. Also, because of the parallel arrangement of the sn-1 and sn-2 chains, ether (alkyl) chains cause tighter packing [143,144]. Ether lipids constitute about 20% of the total phospholipids in mammals, and they can be found in various tissues and organs. Another important packing factor is cholesterol (ergosterols in yeast) and sphingomyelin [145,146], which are abundant in the plasma membrane (PM) and endosomal membranes, making them thicker and less elastic. This property is particularly important for the PM, given that it serves as the boundary between the cell and outside.

The amphipathic-lipid-packing-sensor (ALPS) motif serves as a packing sensor. This motif was first identified in ARFGAP1 [147]. It has the property of forming an amphipathic alpha-helical structure, with its polar face weakly charged mainly by serine and threonine residues [148]. With these Ser/Thr residues, ALPS and ALPS-like motifs recognize membrane curvature [148]. Hydrophobic residues of ALPS motifs are inserted into loosely packed membrane surfaces, but they seem to be insensitive to electrostatics because of the lack of basic residues [148]. Indeed, the ALPS motif in the ATG14L BATS domain (Figure 2A,B) serves as a packing sensor of VPS34 complex I [26]. Packing defects can also be caused by membrane thinning in an artificial membrane system in which the ArfGAP1 ALPS motif is recruited to the thinned area [149].

### 7.3. Membrane Curvature

In the cell, membrane curvature can be generated either spontaneously or by protein domains or motifs (Figure 5B). Spontaneous membrane curvature can be caused by noncylindrical lipids. For instance, because relative volumes of the polar heads of PE, PA, cardiolipin (CL), and diacylglycerol (DAG) are smaller than their acyl chains, these lipids are inverted cone-shaped, generating negative curvature into membranes. Conversely, relative volumes of polar heads in lysophospholipids are larger; therefore, they are cone-shaped, resulting in positive curvature in membranes [150,151,152]. In regard to protein-generated curvature, some amphipathic helices (e.g., alfa-synuclein and Sar1), and C2 domains (e.g., Synaptotagmin) insert into membranes to generate membrane curvature [152,153,154,155]. In endocytic pathways, where the lipid composition is similar to that of PM membranes and more tightly packed and less flexible than that of ER membranes [156,157], curvature is generated by several representative proteins, such as sorting nexin (SNX), and endophilin proteins that have BAR domains [158,159,160,161]. In contrast to the ALPS motif, BAR domains not only sense membrane curvature but generate and enhance it (Figure 5B) [162,163,164]. This is due to the facts that the membrane-associating regions of the BAR domains are composed of basic amino acids and that endocytic membranes are more abundant with PS than ER membranes, in which PS is mostly found on the luminal side [140]. In addition, BAR domain-containing proteins also tend to form oligomers [152,162]. Charge-driven interaction is not in common with the ALPS motif [140]. A recent study also suggested that protein phase separation can cause membrane bending or curvature [165].

### 7.4. Electrostatics from PS

PS is a major phospholipid, accounting for ~10% of total biological membranes [166,167]. The negatively charged property of PS can attract various positively charged protein patches to enhance protein–membrane interactions and can activate various protein/lipid kinases/phosphatases both in vitro and in vivo [168,169,170,171,172]. PS is synthesized in the ER and mitochondria-associated membranes (MAM, Figure 6), from which PS is transferred throughout the cell via vesicle trafficking or PS-specific lipid transfer proteins [167]. The ER and mitochondria have low levels of PS (~4% and ~1%, respectively), and they are rarely colocalized with the PS binding probe, Lact-C2 [173], indicating that PS is localized at the luminal side of the ER. This ER-localized PS is used mainly as a precursor to synthesize phosphatidylethanolamine (PE) [174]. In contrast, PS is asymmetrically enriched in the cytoplasmic leaflet of the plasma membrane (PM); therefore, many PS-binding proteins are also PM-associating proteins [174].

### 7.5. The Use and Interpretation of High-Percentage PS In Vitro

A high concentration of PS could cause an artificial effect in vitro. For example, we recently showed that not only human VPS34 complexes I and II but even VPS34 alone could be greatly activated on giant unilamellar vesicles (GUVs) containing 25% DOPS compared with 10% DOPS-containing GUVs [24]. This condition was purposefully designed to see an exaggerated effect of electrostatics on lipid kinase activities in vitro. As mentioned below (see Section 7.7), PS can be locally enriched in ER membranes, where complex I is recruited during starvation, and has an important role on early endosomes [157], to which complex II and its recruiting protein Rab5 are mainly localized, but it remains elusive whether this activation by 25% DOPS could be physiological or not. In particular, VPS34 alone lacks the membrane-associating adaptor arm (Figure 2B) and was inactive on 10% DOPS-containing GUVs. Nonetheless, it could be activated by high-PS-containing lipids [24]. Therefore, the VPS34 result may be interpreted as an artificial effect. For performing in vitro liposome experiments, it is noteworthy that commercially available lipid substrates are mostly designed for selecting strong inhibitors as part of a drug screen. Therefore, they are designed to maximize activity and made of only PtdIns or simple mixtures of PtdIns and PS at the ratios of 1:9 (PV5122, Thermo Fisher) or 1:3 (V1711, Promega), at which the PS concentrations are extremely high. When researchers plan to measure physiologically relevant lipid kinase activities or membrane binding, it is highly recommended to make liposomes of which the lipid composition mimics that of the organelle of interest. Recently published guidelines described the experimental design in more detail [175]. To examine the effect of PS on the interaction or activation of the protein of interest in vitro, it is also advisable to colocalize the protein of interest with a PS marker or binding probe such as Lact-C2 [176,177] in cells in advance. This may ensure the physiological correlation between the localization of the protein of interest and PS.

### 7.6. Electrostatics from Phosphoinositides

As mentioned in the introduction, there are seven possible phosphoinositides depending on the position and the number of phosphates on the inositol ring. Because of the negatively charged property of the phosphate group(s), phosphoinositides can provide membranes with electrostatics. Similarly to PS, this can attract phosphoinositide-binding proteins with clusters of basic residues such as actin regulatory proteins [178]. Although this may lead to the assumption that the order of binding strength or activity of phosphoinositide-binding proteins should be PtdIns(3,4,5)P_3_ > biphosphates > monophosphates, this was not the case, at least with human VPS34 complexes I and II [24]. It appears that the protein–phosphoinositide interaction might be regulated by either stereochemistry, the net charge of the phospholipid (i.e., the number of phosphates on the inositol ring), or the balance between these factors [179,180,181].

### 7.7. Physicochemical Properties of Autophagosomal Membranes

In mammals, autophagy appears to be initiated at multiple sites on ER membranes [182,183,184,185]. ER membranes are composed of unsaturated lipids with shorter acyl chains and less phosphatidylserine (PS) and cholesterol than plasma membrane lipids, which are composed of more saturated lipids with longer acyl chains and higher levels of PS and sterols [140,186]. This ER lipid property results in flexible and highly curved membranes that may be able to attract various packing-sensing proteins. Indeed, the yeast Atg1 initiation complex prefers highly curved membranes to bind to [97,187]. In particular, highly curved membrane structures called omegasomes are a hallmark of initiation/nucleation of autophagy in mammalian cells (Figure 6) [183]. It is not clear whether omegasome formation is promoted by ATG proteins or ATG proteins use the already formed omegasome. The omegasome is marked by the PtdIns(3)P-binding protein DFCP1, indicating that complex I and PtdIns(3)P effectors prefer highly curved membranes to associate with (also see Section 8). As mentioned above, PS is not abundant in ER membranes. However, during starvation, phosphatidylserine synthase 1 (PSS1), which synthesizes PS from phosphatidylcholine (PC), is colocalized with FIP200 independently of PtdIns(3)P [53]. PSS1 is localized mainly to MAM [188], wherein ATG14L was also found to be recruited during starvation (Figure 6) [189]. This indicates that even if the average PS concentrations in the cytoplasmic side of the ER and mitochondria are low, it may be possible to increase the local PS concentration in a continuous membrane during starvation. Although it remains to be seen, this locally increased PS may be used to attract various proteins such as the ULK1/Atg1 complex to initiate autophagy. Taken together, these ER membrane properties—flexible, highly curved, and locally enriched with PS during starvation—may constitute an ideal platform to initiate autophagy. After initiation, further membrane curvature is generated. Although the protein identities for this role have not been revealed in mammals [190], in yeast, the sorting nexin Atg20–Atg24 (Snx4) heterodimer, which is known to be on the PAS [191] and is downstream of Vps34 [192], can cause membrane tubulation [193]. Also, a lipidated form of the yeast Atg8 was recently found to have a membrane deformation role [194] (Figure 6). These findings suggest that the physicochemical properties of membranes are important determinants for the formation of the autophagosome [195]. The following examples of in vitro studies indicate that three physicochemical properties—lipid saturation, packing, and electrostatics—greatly affect the activities and membrane binding of ATG/Atg proteins and VPS34 complex I, as summarized in Table 1.

### 7.8. Effects of Lipid Packing, Curvature, and Electrostatics on the Activity and Membrane Binding of VPS34 Complexes

Because VPS34/Vps34 is a lipid kinase, lipid environments greatly affect the activity and membrane binding of VPS34/Vps34 complexes. As briefly mentioned in Section 3, this phenomenon is mediated by specific motifs in the complexes. The membrane binding and activity of human complex I depends on the ALPS motif in the BATS domain of ATG14L [24,26,28,140]. The ATG14L ALPS motif senses membrane curvature, with preference for small vesicles or high membrane curvature [26]. It must be noted that yeast Atg14 does not have the BATS domain; therefore, yeast complex I activity is much lower than yeast complex II activity in vitro on flat membranes [23]. Nonetheless, yeast Atg14 shows clear PAS localization in the cell [207,208]. It remains unclear how the yeast complex I is recruited to and becomes activated on the PAS. The membrane binding and activity of human complex II depend on two aromatic finger motifs (aromatic fingers 1 and 2) and the hydrophobic loop, all of which are in the BARA domain of Beclin 1 [24,30] (Figure 2B). In the yeast complex II, only the aromatic finger 1 motif was found to be responsible for membrane binding and activity [23]. The activities of both human complexes are greatly influenced by membrane packing. With increasing lipid saturation, both human complexes become less active [24]. This packing effect was found both with nonsubstrate lipids and with the substrate PtdIns [24]. Interestingly, this packing effect was partially compensated by membrane curvature, i.e., the complexes still showed some activities on smaller vesicles composed of half-saturated lipids [24]. Also as mentioned above, electrostatics represented by PS greatly influence the activity of both complexes, although experiments on this topic need to be carefully planned and interpreted.

### 7.9. Effects of Packing, Membrane Curvature, and Electrostatics on Membrane Tethering (MT) and Lipid Transfer (LT) of ATG2A/B/Atg2

Within 30 min after starvation induction, autophagosomes with sizes around 400–900 nm/500–1500 nm (yeast/mammals) are generated to engulf cargos [209,210]. This means that large amounts of lipids have to be promptly provided to form the double-layered autophagosomal membranes. Lipid-transfer proteins ATG2A and B in mammals and Atg2 in yeast regulate this process at the downstream of complex I [79,81,211]. Recent cellular, in vitro, and structural studies have revealed that these proteins have activities of membrane tethering (MT) and lipid transfer (LT) bipartitely, so that lipids can be transferred from the membrane source(s), which is/are likely the ER or ATG9/Atg9 vesicles, to the isolation membrane or phagophore [197,198,199,212,213,214]. This process is facilitated by the VPS34/Vps34 effector proteins, WIPIs/Atg18 [200,201,202,211,215]. ATG2A/B has the strongest affinity to WIPI4 [216], whereas the Atg2–Atg18 interaction is weak [202]. because of their involvement in LT, the effects of the three physicochemical parameters—packing, membrane curvature, and electrostatics—on MT and LT activities have been intensively characterized (Table 1). The yeast Atg2 is known to recognize packing defects [199]. Also, the LT of ATG2A–WIPI4 is more efficient with small unilamellar vesicles (SUVs) composed of DO (18:1–18:1) lipids than with PO (16:0–18:1) SUVs [200]. All of ATG2A/B/Atg2 prefer smaller vesicles for their membrane binding, MT, and LT [200,202,215]. High PS concentrations enhance the membrane binding, MT, and LT of ATG2A [200] and the MT and LT of ATG2B [202]. The MT and LT of ATG2B can be also enhanced by PtdIns(3)P [202]. This high PS can bypass the necessity of WIPI4 for the MT and LT of ATG2A/B [200,216]. On the other hand, the MT and LT of the yeast Atg2 in the absence of Atg18 (see below) is decreased by high PS and PtdIns(3)P [202], indicating that while their MT and LT activities are conserved through evolution, detailed mechanisms might be differently regulated. Notably, the yeast Atg2 without Atg18 can activate MT and LT in vitro [202,215], whereas in the cell, functional Atg18 is required for the localization of Atg2 to autophagic membranes [79,217]. It remains to be seen which step or pathway these in vitro MT/LT activities of Atg2 without Atg18 reflect.

### 7.10. Roles of Lipid Unpacking and Curvature on the Atg16/ATG16L Complex and the Lipidation of Atg8 Families

The ubiquitin-like Atg8 family proteins (Atg8 in yeast and LC3/GABARAP proteins in mammals) are among the core ATG/Atg proteins. Unlike ubiquitins, they are lipidated by PE, and these lipidated forms are essential for autophagosome biogenesis. The human ATG16L complex, which is an E3-like enzyme composed of ATG12-ATG5-ATG16L, can specify the LC3 lipidation membrane for autophagosome biogenesis by recruiting LC3 [218]. This lipidation is dependent on the E2-like ATG3/Atg3. In yeast, the PAS localization of the Atg16 complex and Atg3 is Vps34 complex I-dependent [219], indicating that PtdIns(3)P synthesis on the PAS is essential for the recruitment of downstream factors [79,80]. In vitro studies suggested that LC3 recruitment to the membrane and its lipidation are also greatly affected by lipid physicochemical parameters. On SUVs, yeast Atg8 lipidation occurs only with Atg12–Atg5, whereas on GUVs, Atg16 is required for efficient Atg8 lipidation [204]. Furthermore, the LC3B lipidation by the mammalian ATG16 complex occurs more efficiently on GUVs composed of DO (18:1–18:1) lipids than on GUVs composed of PO (16:0–18:1) lipids [203], indicating that both membrane curvature (vesicle size) and packing are critical factors for the lipidation of Atg8 family proteins. The lipidated forms of the human Atg8 family proteins GATE-16 and GABARAP promote membrane tethering and fusion. In particular, vesicle diameter and negative intrinsic curvature-inducing lipids (cardiolipin (CL) and diacylglycerol (DAG)) (Figure 4B) facilitate fusion events [205]. Even among the human Atg8 family proteins, it appears that each protein has its own curvature preference, since LC3B can tether SUVs more efficiently than GATE-16, whereas GATE-16 can tether LUVs more efficiently than LC3B [206]. In line with cell biological observations [220,221], these in vitro findings suggest that each of the human Atg8 family proteins may have a unique role at a different stage of autophagosome biogenesis by targeting a membrane with a different physicochemical property. Furthermore, a recent study revealed that the physiologically lipidated form of yeast Atg8 could induce positive membrane curvature in vitro [194].

### 7.11. Phosphatidylinositol (PtdIns), the Sole Substrate for VPS34/Vps34

PtdIns makes up 10–15% of the total lipids in mammalian cells [222]. Because VPS34/Vps34 uses PtdIns as a sole substrate, the localization of PtdIns is equally important as the intracellular localization of VPS34 complexes. For example, although the cytoplasmic leaflet of the plasma membrane (PM) is abundant with PS [174], which greatly activates both complexes I and II (see Section 7.4) [24], PtdIns(3)P is rarely observed there. This could be partly due to very low levels of PtdIns at the PM (see below) [223,224] and tight lipid packing in PM membranes [140,174], where VPS34/Vps34 are not active [24] (see Section 7.8).

PtdIns is synthesized by PI synthase (PIS) from cytidine diphosphate-diacylglycerol (CDP-DG) and inositol in the ER (Figure 6) [225]. Except for plants, most eukaryotes express a single PIS gene [225,226]. PIS is an ER-localized transmembrane protein. Upon starvation, the ULK1 complex is recruited to PIS-positive structures, which do not colocalize with ERGIC and ER exit markers or mitochondria [53]. The PtdIns synthesis at the PIS-positive structures is necessary for the recruitment of the ULK1 complex and the following formation of the isolation membrane [53]. This indicates that PtdIns has a greater role in the formation of autophagosomes than as a substrate for VPS34. In yeast, the relative PtdIns content in whole cell membranes was increased by starvation from 12 to 46% [227]. Also, acyl chain profiles of phospholipids were shifted to longer and more unsaturated [227]. This indicates that during starvation, intracellular phospholipid composition can be dramatically rearranged. The same study also measured the phospholipid composition of autophagosomes by immunoisolating Atg8-containing vesicles and found 37% of PtdIns. Also, around 60% of phospholipids had two unsaturated acyl chains [227]. These results indicate that the high PtdIns content and unsaturated acyl chain property are key features of yeast autophagosomes. The lipid composition in autophagic membranes might be different in organisms. In *Drosophila* autophagosomes that were immunoisolated by Atg8a, the PtdIns concentration was around 10%, and phosphatidylethanolamine (PE) was the major component in the wildtype animals, whereas the PtdIns concentration was dramatically increased in autophagosomes from *Atg2*^−^ flies [228]. Similarly to the case of yeast, over 60% of *Drosophila* autophagic membranes were unsaturated [228]. The high abundance of unsaturated lipids is a common feature of autophagosomal membranes at least between yeast and flies, although this approach cannot exclude the possibility that measurement of the lipid compositions also includes engulfed membranes inside autophagosomes.

PtdIns is also known to be abundant in yeast Atg9 vesicles, in which PtdIns makes up over 40% of the vesicle lipids [197]. Both in yeast and mammals, ATG9/Atg9 vesicles are 30–60 nm in diameter [81,196] (Table 1) and derived from the Golgi [196,229,230,231]. Because human ATG9 vesicles do not contain PIS [53], the PtdIns in ATG9 vesicles might be synthesized elsewhere. Since PtdIns is enriched in the Golgi (see below), it appears that ATG9 vesicles might bud off from PtdIns-concentrated regions in Golgi membranes, although the mechanism for this remains to be seen. Human complex I has a binding preference for PtdIns compared with complex II [28]. As mentioned above, human complex I is more reactive to unsaturated PtdIns than saturated PtdIns, and both human complexes I and II are more reactive to membranes composed of unsaturated nonsubstrate lipids (also see above) [24,232]. Collectively, high-PtdIns-containing, highly curved (small), and unsaturated (flexible) membranes with Rab1/Ypt1 (see Section 3.2) of ATG9/Atg9 vesicles can be ideal platforms for complex I to activate during autophagy.

Except for the autophagosome and the ER, from which PtdIns is distributed throughout the cell, exact intracellular localization of PtdIns has been enigmatic. While subcellular fractionation followed by lipid mass spectrometry can provide quantitative information on lipid species, including acyl chain length and saturation, it lacks the topology information. Also, because organelle membranes are to some extent interconnective, it is technically difficult to resolve each organelle fraction and to fractionate membrane contact sites between organelles. Recently developed methods from two groups to detect intracellular PtdIns may complement these approaches [223,224]. These include a PtdIns-specific binding probe, *Bc*PI-PLC^H82A^, and indirect monitoring of diacylglycerol (DAG), a converted product from PtdIns by PI-PLC on specified organelles. These studies agreed well on high PtdIns abundance in the Golgi and low abundance in the PM [223,224]. Interestingly, although PtdIns(3)P is well known to be enriched in early endosomes, in which an early endosomal small GTPase Rab5 recruits complex II [31,43,44,45] (Figure 3B, also see Section 8), *Bc*PI-PLC^H82A^ could not be found in either early or late endosomes [223]. Endosomal PtdIns(3)P production was greatly affected by the ER-residing PtdIns [223], indicating that the endosomal PtdIns could potentially be provided from the ER rather than through de novo synthesis at endosomes. This may be achieved by the direct connection between the ER and endosomes [233,234]. An alternative hypothesis could be that PtdIns-specific lipid transfer proteins (LTPs) may directly provide PtdIns to complex II [235], although this is unlikely given that the heterotetramer complex II requires a membrane platform to activate [23,24]. Although *Bc*PI-PLC^H82A^ could not detect PtdIns in late endosomes, there was an indication of PtdIns existing there [223]. Because endosomes are enriched with phosphoinositide phosphatases [236], one possibility is that some monophosphoinositides might be quickly converted back to PtdIns alongside direct PtdIns transfer from the ER (also see Section 8.3).

### 7.12. Membrane Properties in Health and Disease

In actual human bodies, lipid compositions differ between sexes and between aged and young populations [237]. For example, alkyl (ether) and saturated forms of phospholipids are increased in older populations compared with younger populations [237]. Also, changes in lipid compositions have been found in various diseases, including a rare neurometabolic syndrome (Sjögren–Larsson syndrome (SLS)) [238] and various types of cancer [239]. Changes in PtdIns properties have been associated with human health and diseases. In mammalian cells, PtdIns is selectively enriched with stearic acid (C18:0) at the sn-1 position and arachidonic acid (C20:4) at the sn-2 position, designated as 38:4 or SAPI. This occupies up to 30–85% of total PtdIns species depending on the tissue and cell type [225]. A metastasis-promoting mutation in p53 (*Trp53^R172H^*) [240] increased the level of 18:1–18:1 PtdIns (dioleoyl PI, or DOPI) in mouse pancreas and MEFs [241]. Also, an increased level of the wildtype p53 was associated with a shift in phospholipids, especially PtdIns, from having two monounsaturated acyl chains to having one or no monounsaturated acyl chains [242]. These studies indicate that p53 may repress the unsaturation of acyl chains, which could be an important part of supressing tumorigenesis. Human VPS34 complex I is activated more strongly by DOPI than SAPI [24]. Therefore, it is likely that in p53-dysfunctional cells, VPS34 complexes might be more active. In aggressive cancerous prostate tissues, an increased proportion of PtdIns with 0–2 double bond-containing acyl chains in relation to that with ≥3 double bond-containing acyl chains was found [243]. Also, in B16 cells, 38:4 PtdIns was greatly reduced by heat shock and benzyl alcohol (BA), a membrane fluidizer [244]. These findings indicate that even if the genes encoding VPS34 complex subunits are not mutated, the activity of the complexes can be greatly influenced by surrounding environments that affect lipid compositions.

## 8. VPS34 Activation Regulation by Feedback Mechanisms and Downstream Effectors

### 8.1. Autophagy

The PtdIns(3)P produced by complex I during autophagy recruits downstream effectors such as WIPIs and DFCP1, which leads to nucleation of phagophores [80,245] (Figure 6). The WD-repeat proteins interacting with phosphoinositides (WIPIs) or PROPPINs are the only known PtdIns(3)P-binding effectors that are essential for autophagy and conserved from yeast to mammals. Humans have four WIPIs (WIPI1 to WIPI4), and yeasts have three, Atg18, Atg21, and Hsv2. They fold as seven-bladed β-propellers [246,247,248] and bind to both PtdIns(3)P and PtdIns(3,5)P_2_ with blades 5 and 6 [247]. For the membrane localization of WIPIs, PtdIns(3)P is essential for both yeast and mammals [80,217]. WIPIs bind to ATG2A/Atg2 [200,201,216,217,249] (see Section 7.9) and ATG16 L1 [203,250] separately. In each case, the WIPI facilitates or enhances the function of the binding protein or protein complex [200,203]. Fracchiolla et al. reported that LC3B lipidation by the ATG12–ATG5–ATG16L1 complex was enhanced by WIPI2d. Furthermore, the same study found that VPS34 complex I activity was enhanced by WIPI2d, which led to positive feedback on PtdIns(3)P synthesis [203] (Figure 7A). Interestingly, Birgisdottir et al. found several LC3 interacting regions (LIRs) in the subunits of human VPS34 complex I, and the Atg8 family proteins GABARAP and GABARAPL1 preferably bound to VPS34 complex I [251]. Although the consequence of the interaction between the Atg8 family proteins and VPS34 complex I remains to be seen, one may hypothesize that there could be another positive feedback loop between VPS34 complex I and Atg8 family proteins to enhance PtdIns(3)P synthesis and the lipidation of Atg8 family proteins (Figure 7A).

### 8.2. Endocytic Pathway

Unlike autophagy, the endocytic pathway is branched into multiple pathways with various membranous compartments. These endosomal compartments are sorted in a variety of ways, and their membranes undergo a variety of membrane trafficking and deformations, including tubulation, fission/scission, tethering, and fusion. These are vital for receptor and ion channel recycling, formation of multivesicular bodies (MBVs), and endosome maturation from early to late endosomes [252,253,254]. PtdIns(3)P is essential for the majority of these events. Also, while much receptor-mediated signal transduction takes place at the plasma membrane, early endosomes can also be a platform for some signalling molecules.

As described in Section 3.2, complex II is recruited and activated by Rab5 on early endosomes [31,44,45,255], where PtdIns(3)P is produced and used as a signalling molecule for downstream effectors (Figure 3B). The activity and patterns of cellular localization of small GTPases are coupled with their interactions with GEFs and effectors. In the case of Rab5, the Rabex5–Rabaptin5 complex is its GEF-effector complex [156,256,257], and this Rab5–GEF-effector interaction enhances the domain formation on membranes [156]. This domain formation is further enhanced by PtdIns(3)P, indicating the presence of a positive feedback loop between PtdIns(3)P synthesis and the Rab5–GEF-effector complex, potentially via PtdIns(3)P binding to the hypervariable region of Rab5 [258]. The locally concentrated Rab5 may recruit its effectors to activate their downstream pathways. This positive feedback mechanism is reminiscent of the positive feedback in the complex I–WIPIs–ATG16 complex during autophagy (Figure 7).

### 8.3. PtdIns(3)P-Binding Proteins in Endocytic Pathways

In endocytic pathways, there are a variety of PtdIns(3)P-binding proteins that typically carry the PX or FYVE domains [259,260,261]. Although more protein domains have been reported to bind to PtdIns(3)P [262], I focus on these two domains. While both of them have a binding specificity for PtdIns(3)P, their structures and PtdIns(3)P binding mechanisms are different [263]. For example, while the PtdIns(3)P binding of the FYVE domains is pH sensitive [264,265], PX domains bind to PtdIns(3)P independently of pH [265].

A representative protein family for PX domain-carrying proteins is the large sorting nexins (SNXs) family [266]. They can be found as parts of protein complexes such as the retromer and retriever complexes that are involved in recycling receptor proteins [266,267]. In addition, many SNXs have BAR domains that are involved in membrane tubulation/deformation (see Section 7.3). Therefore, the PX-BAR-containing SNXs deform membranes into tubules in a PtdIns(3)P-dependent manner [268]. The PtdIns(3)P-binding PX domains have been also found in other proteins, including a SNARE protein, Vam7/VAMP7 (yeast/mammals), which is involved in membrane fusion [269]; phospholipase Ds (PLDs) that produce phosphatidic acid (PA) and soluble choline from phosphatidylcholine [270] (also see Section 5.2 and Figure 4B); and pPhoX40, a member of the Phox complex, a phagocytic NADPH oxidase complex [271]. Another enzyme with a PX domain is the serum and glucocorticoid-regulated kinase3 (SGK3), a serine/threonine kinase that acts at the downstream of the class I PI3K–PDK1 signalling independently of AKT [272,273]. The PtdIns(3)P binding via its PX domain is important for the activation SGK3 itself [274,275]. The existence of these PtdIns(3)P-binding PX domains in a wide range of intracellular pathways reflects the importance of VPS34 and PtdIns(3)P for cellular homeostasis.

The FYVE domain was named after the proteins Fab1, YOTB, Vac1, and EEA1. Compared with PX domains, it appears that FYVE domains are more specialized for PtdIns(3)P binding [260]. FYVE-containing proteins tend to homodimerize, which enhances the affinity to PtdIns(3)P [260]. By using this characteristic, the Stenmark laboratory developed the 2xFYVE reporter, wherein two mouse HRS FYVE domains were tandemly arranged. This greatly enhanced the PtdIns(3)P binding compared with the single FYVE domain and has been widely used for monitoring intracellular PtdIns(3)P localization [255]. Similarly to PX domains, FYVE domain-containing proteins can be found in a variety of cellular events. EEA1 is a Rab5 effector and tethers homotypic early endosomes [276,277,278]. As mentioned above, VPS34 complex II is also a Rab5 effector, indicating that Rab5 serves as a hub wherein its effectors and effector’s product further interact to enhance downstream events (Figure 7B). Other protein examples that contain the FYVE domain include RUFY family proteins, DFCP1, and SARA. The recently redefined RUFY (Run and FYVE domain-containing) family proteins are found in endocytic pathways, phagocytosis, and autophagy [279]. DFCP1 is found in the cis-Golgi during rich-medium conditions, then translocated to the omegasome, the hallmark structure of autophagosome initiation/nucleation (Figure 6) [183]. Smad anchor for receptor activation (SARA) is involved in TGF-beta-induced Smad2 nuclear localization [280,281]. Importantly, FYVE domains are also found in phosphatidylinositol-3 phosphatases (MTMRs), which dephosphorylate PtdIns(3)P to synthesize PtdIns (also see Section 4), and in phosphatidylinositol 3-phosphate 5-kinase Fab1/PIKFyve (yeast/mammals), which phosphorylate the 5′-OH position of PtdIns(3)P to synthesize PtdIns(3,5)P_2_ [282,283]. This indicates that unlike that in autophagy, PtdIns(3)P in endocytic pathways is used as a precursor for PtdIns and PtdIns(3,5)P_2_. Because MTMRs and Fab1/PIKFyve are found mainly in late endosomes, PtdIns(3)P is less abundant there than in early endosomes, although it is still essential for the maintenance of late endosomes, since the lack of VPS34/Vps34 causes a loss of multivesicular bodies (MVBs) [255,284,285] (also see Section 7.11).

## 9. Regulation of VPS34 Activation by Binding Proteins

As mentioned above, interactions with Rab1a and Rab5a greatly increase the activities of complex I and complex II, respectively [31,45]. Other well-known VPS34 complex-binding proteins are the fifth subunits of the complexes, NRBF2/Atg38 (mammals/yeast) for complex I and Rubicon for complex II. Unlike the Rab proteins, the interactions of which with the complexes are very weak and transient, the interactions between the fifth subunit and the complex are strong, and the pentameric complexes can be purified through size exclusion chromatography [29,48,55,114]. On the other hand, their roles in the activity of the complexes are controversial. While it is agreed that the complex I-specific NRBF2/Atg38 is found on the autophagosome, it has been reported both as a positive and a negative regulator of complex I (see review [34]; also see Section 5). Also, the exact binding subunit(s) of complex I for NRBF2/Atg38 and binding stoichiometry between complex I and NRBF2/Atg38 remain to be elucidated [34]. Autophagy and in vitro studies have agreed that Rubicon is a negative regulator of autophagy and complex II [20,29]. However, during LC3-associated phagocytosis (LAP), a noncanonical autophagy, Rubicon acts as a positive regulator of complex II [286]. Rubicon is also known to bind to Rab7 via its C-terminal Rubicon homology (RH) domain [287,288], whereas it has been also reported that Rab7 also binds to VPS15 [289]. Another study showed that Rab7 competed with UVRAG, the complex II-specific subunit for the Rubicon binding [290]. It remains to be seen how these Rab7 interactions with Rubicon and VPS15 (potentially complex II) take place.

In addition, a variety of proteins, including Ambra1 [291], PAQR3 [292], GORASP2/GRASP55 [293], PDPK1 [294], and SH3GLB1/Bif-1 [295], have been found to interact with the VPS34 complex subunits. Because most of these interactions were examined by immunoprecipitation or yeast two-hybrid assays, most of their direct interactions with and direct involvements in the activation of VPS34 complexes remain to be seen. For example, although Ambra1 was found as a Beclin 1-interacting protein in 2007 [291], the complex specificity of Ambra1 has been still elusive. Also, Ambra1 pulldown could not detect Sypro Ruby-visible Beclin 1 [296], indicating that the interaction was very weak or indirect. In contrast, GAPR-1/GLIPR2, which was found as an interacting protein of the Tat-beclin1 peptide [297,298] (see below), was recently found to bind to purified VPS34 complex I, negatively regulate VPS34 complex I activity, and serve as a negative regulator of autophagy [298]. Also, it has been reported that Beclin 1 binds to the HIV-1 Nef, an antiautophagic maturation protein, via a region containing the β1-β2 ß1–ß2 strands in the Beclin 1 BARA domain [297,299]. In addition, an in vitro study showed that HIV-1 Nef inhibited complex II activity [29]. By using this HIV-1 Nef–Beclin 1 interaction, a fusion peptide between a Beclin 1 BARA domain fragment and a cell-permeable peptide derived from the HIV-1 Tat protein transduction domain, the Tat-beclin1 peptide, was designed as an autophagy activator and indeed increased autophagic activity [297]. Because VPS34 and Beclin 1 are shared between complexes I and II, inhibitors or activators targeting these subunits affect the activity of both complexes. To overcome this, Wu et al. engineered the coiled-coil domain of UVRAG to enhance its binding to the Beclin 1 coiled-coil domain and to outcompete with ATG14L and Beclin 1 self-dimerization [300]. The same study also designed a “stapled” peptide to promote the interaction between the Beclin 1 coiled-coil domain and the coiled-coil domain of UVRAG or ATG14L as well as to reduce Beclin 1 dimerization [300]. Finally, although it is not a protein, a screen to find small molecules that selectively inhibit the formation of complex I was recently developed and resulted in the successful discovery of a compound [301].

## 10. Concluding Remarks and Future Perspectives

Recent progress in cryo-EM and cryotomography, and in other biophysical techniques such as HDX-MS and single molecule kinetics, has helped understand the detailed activation mechanisms of VPS34/Vps34 complexes. These approaches have revealed that the kinase domain of VPS34/Vps34 is not the sole activation determinant of activity. Instead, activity is greatly influenced by the binding subunits of the complexes, lipid environments, and accessory regulatory proteins. Therefore, in some human diseases, where the above influencing factors are altered, it might be possible that the VPS34 activity is significantly altered even though the genes encoding the VPS34 complex subunits are not mutated. To examine this, developing a quantitative, physiologically relevant, and handy method to measure VPS34 activity in vivo as well as in vitro is essential. In the cell, since the PtdIns(3)P-driven signalling in autophagy and endocytic pathways is regulated spatially and temporally, it is conceivable that VPS34/Vps34 complexes may interact with their binding proteins and membranes only transiently, as the cases of the complex I–Rab1a and complex II–Rab5a interactions. Methods to precisely screen these weak and transient interactions such as BioID will be powerful tools for the future proteomics of VPS34 complexes. Lastly, this knowledge on activation mechanisms can be applied to translational efforts to develop more specific activators and inhibitors for the treatment of human diseases.

## Figures and Tables

**Figure 1 cells-10-03124-f001:**
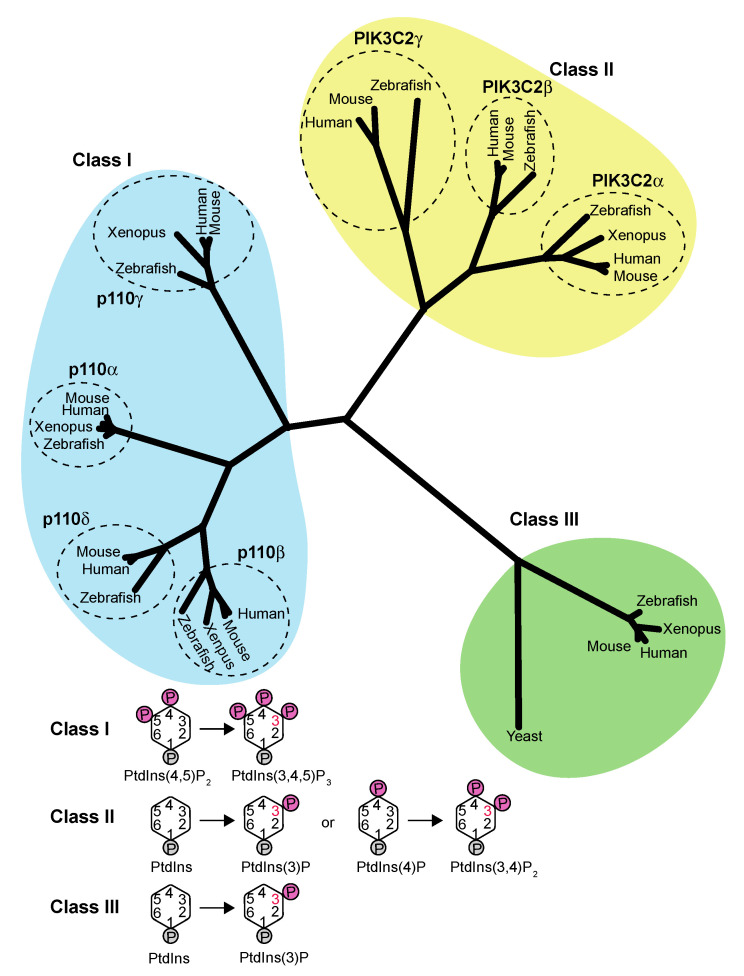
The class III PI3K VPS34/Vps34 is the most ancestral PI3K among the three classes of PI3Ks. The Class I PI3Ks phosphorylate PtdIns(4,5)P_2_ to synthesize PtdIns(3,4,5)P_3_, while the Class II PI3Ks phosphorylate PtdIns or PtdIns(4)P to synthesize PtdIns(3)P or PtdIns(3,4)P_2_, respectively. The Class III PI3K (PIK3C3 or VPS34/Vps34 for metazoans/yeast) phosphorylates PtdIns to synthesize PtdIns(3)P. This phylogenetic tree was generated using MEGA (https://www.megasoftware.net/, accessed on 4 March 2021). The following proteins, listed with their Uniprot entry numbers, were used for the analysis: p110α (Human: P42336; Mouse: P42337; Zebrafish: F1QAD7; Xenopus: F6VXG1); p110β (Human: P42338; Mouse: Q8BTI9; Zebrafish: E7F251); p110γ (Human: O00329; Mouse: O35904; Zebrafish: F1RB17); p110δ (Human:P48736; Mouse: Q9JHG7; Zebrafish: Q6NTI2; Xenopus: A0A6I8RLG9); PIK3C2α (Human: O00443; Mouse: Q61194; Zebrafish: F6NMW9; Xenopus: B5DE87); PIK3C2β (Human: O00750; Mouse: E9QAN8; Zebrafish: F1QWN6); PIK3C2γ (Human: O75747; Mouse: O70167; Zebrafish: A0A0G2L6J0); and PIK3C3/VPS34 (Human: Q8NEB9; Mouse: Q6PF93; Zebrafish: F1Q9F3; Xenopus: F6ZM84; Yeast Vps34: P22543).

**Figure 2 cells-10-03124-f002:**
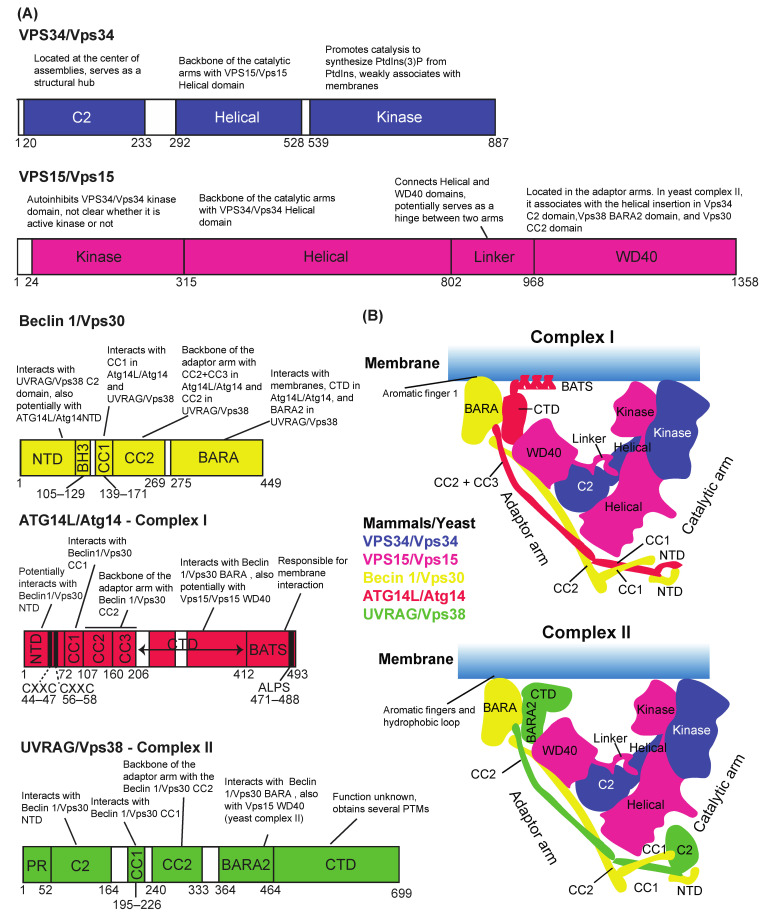
The role of each domain of VPS34/Vps34 complex I and II subunits in the assembly of the complexes. (**A**) Schematic representations of subunits in human numbers. (**B**) Structural schematic representations of complexes I and II. C2: C2 domain; CC: coiled-coil; BARA: β-α repeated, autophagy-specific (BARA) domain; BATS: Barkor/Atg14(L) autophagosome targeting sequence (BATS) domain; NTD: N terminal domain; CTD: C terminal domain; CXXCs: CXXC motifs; WD40: WD40 domain; PR: proline-rich domain; BH3: BH3 domain. The figures are modified from [34,35].

**Figure 3 cells-10-03124-f003:**
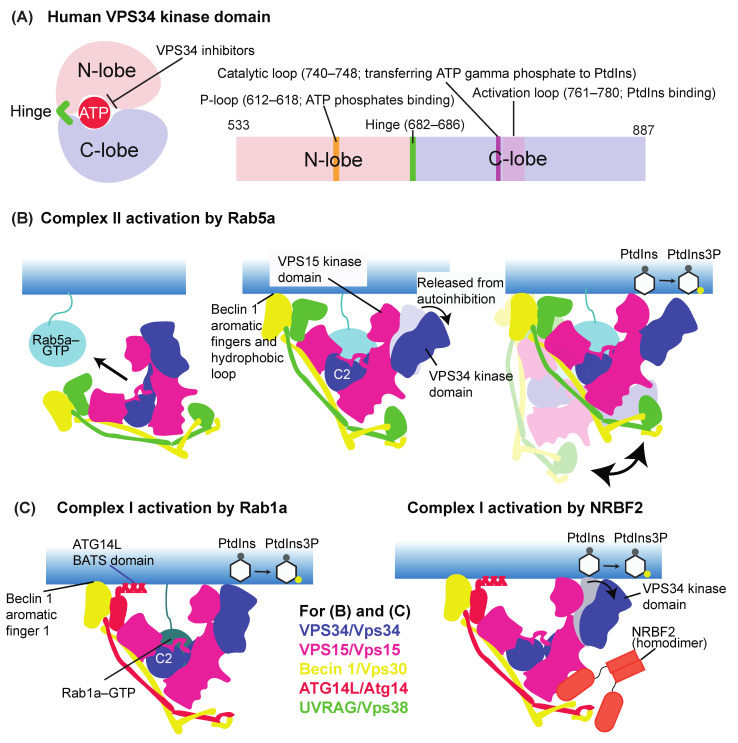
Activation mechanisms of VPS34/Vps34 complexes. (**A**) Schematic representations of the VPS34/Vp34 kinase domain. Left: A simplified structure of the kinase domain. An ATP molecule is sandwiched by the N- and C-lobes, which are connected by the hinge region. Right: A schematic representation of the kinase domain in human VPS34, modified from [35]. Residue numbering shown is for the human VPS34. (**B**) The activation mechanism of human complex II by Rab5a. Left: Complex II is recruited in an autoinhibition form to early endosomes by the early endosome specific small GTPase Rab5a in a GTP-dependent manner. Middle: The Rab5a binding not only recruits complex II to the membrane but releases the VPS34 kinase domain from autoinhibition by the VPS15 kinase domain. Rab5a binds mainly to the C2 helical hairpin insertion (C2HH) of VPS34, along with the small globular domain (SGD) and WD40 domain in VPS15. Along with Rab5a, three motifs in the Beclin 1 BARA domain are responsible for the membrane binding. Right: The membrane-anchored active complex II can tilt up and down to phosphorylate PtdIns and synthesize PtdIns(3)P. (**C**) Activation mechanisms of complex I. In addition to the Beclin 1 aromatic finger 1, the ATG14L BATS domain is essential for the membrane anchoring of complex I. Left: Complex I activation by Rab1a. GTP-bound Rab1a binds to the same C2HH in VPS34 as Rab5a–complex II, but not to the VPS15 SGD or WD40. Right: Complex I activation by the complex I-specific binding protein NRBF2.

**Figure 4 cells-10-03124-f004:**
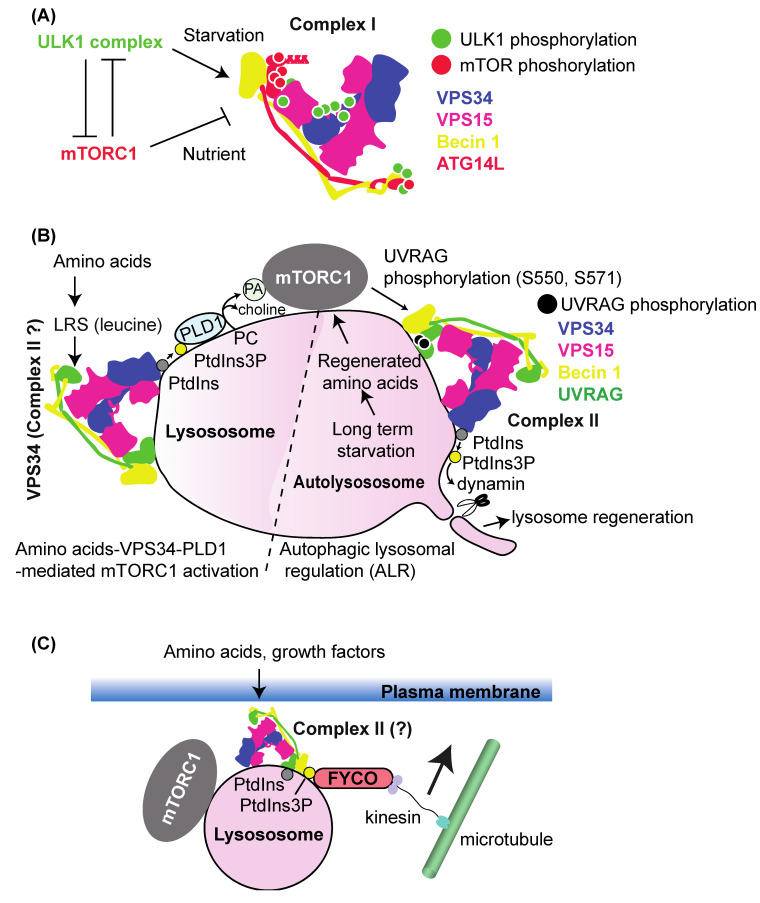
VPS34 complex regulation by the ULK1 complex and mTORC1. (**A**) The ULK1 complex acts as a positive regulator of complex I during starvation, whereas mTORC1 acts as a negative regulator in nutrient-replete conditions. Their phosphorylation positions are indicated in the schematic structure of complex I. Green dots: activating phosphorylation by the ULK1 complex; red dots: inhibiting phosphorylation by mTORC1. (**B**) Left: VPS34 (or, putatively, complex II) acts as a positive regulator of mTORC1 in amino acid-replete conditions. Amino acids activate VPS34 to synthesize PtdIns(3)P, which activates PLD1. Phosphatidic acid (PA), the PLD1 product, binds to mTOR to activate it. Right: After long-term starvation, lysosomes are regenerated by autophagic lysosomal regulation (ALR). Then, accumulated autophagy substrates in autolysosomes are degraded, and amino acids are regenerated. This activates mTORC1 to phosphorylate UVRAG, the complex II-specific subunit, and activate complex II. This activity is necessary for the scission step of ALR, in which the tubulated autolysosome is detached by dynamin. (**C**) VPS34–mTORC1 activation by lysosome translocation. VPS34 (potentially complex II) is activated by amino acids to synthesize PtdIns(3)P, which is bound by FYCO1 via its FYVE domain. This also causes lysosome translocation to the cell periphery via the microtubule–kinesin–FYCO1 interaction. At the plasma membrane, mTORC1 is in close proximity to nutrient-signalling complexes to become activated.

**Figure 5 cells-10-03124-f005:**
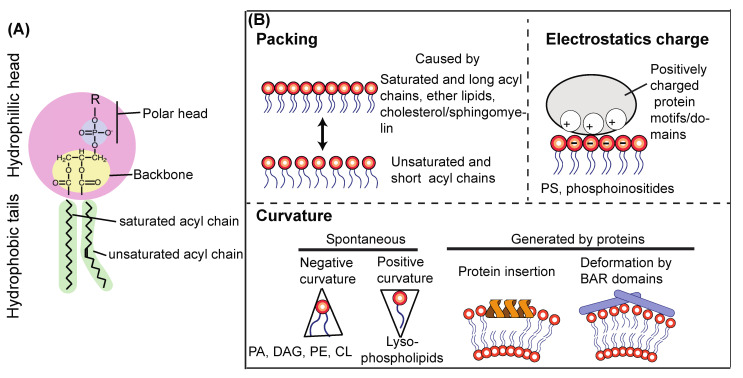
Physicochemical parameters that affect activities and membrane binding of autophagy-related proteins. (**A**) A schematic structure of a phospholipid molecule. A phospholipid is composed of a polar head comprising a modified alcohol (R) and a phosphate, a backbone (glycerol here, also can be sphingosine), and fatty acids or acyl chains. R in the polar head defines the lipid species, such as serine for PS, ethanolamine for PE, choline for PC, and inositol for PtdIns. The polar head and backbone constitute hydrophilic head, facing the aqueous side of the membrane, whereas the acyl chains provide a hydrophobic barrier. (**B**) Three important physicochemical parameters that affect activities and membrane associations of autophagy-related proteins: packing, membrane curvature, and electrostatics.

**Figure 6 cells-10-03124-f006:**
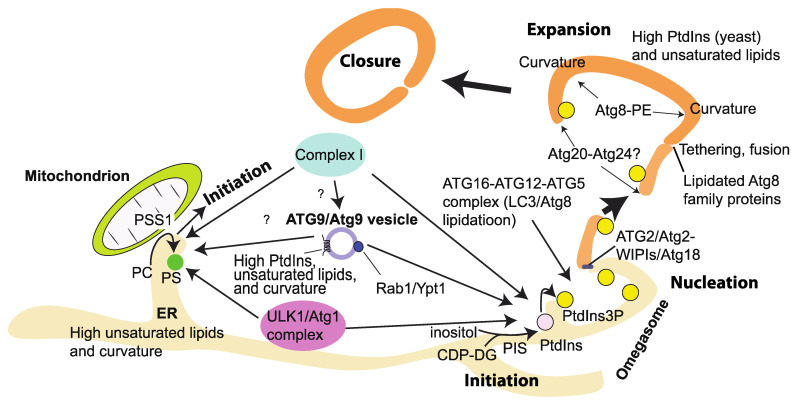
Summary of membrane/lipid environments that are targeted by autophagy-related proteins. During starvation, PtdIns is synthesized in the ER by PIS, which is targeted by the ULK1 complex and essential for the generation of the autophagosome. Complex I, the ATG16–ATG12–ATG5 complex, and the ATG2/Atg2–WIPI4/Atg18 complex prefer unsaturated (unpacked) and highly curved membranes for their activations. ER membranes and ATG9/Atg9 vesicles fulfil these conditions. During expansion, phagophores are tethered and fused, which is facilitated by Atg8 family proteins. Also, curvatures are generated at least by the lipidated yeast Atg8, and the yeast Atg20–Atg24 complex might detect and stabilize membrane curvatures at the edge of the expanding phagophore to support expansion. Along with this, upon starvation, PS synthesis occurs at the ER–mitochondria contact site by PSS1, which is targeted by the ULK1 and complex I. This may also lead to the initiation of autophagosome formation.

**Figure 7 cells-10-03124-f007:**
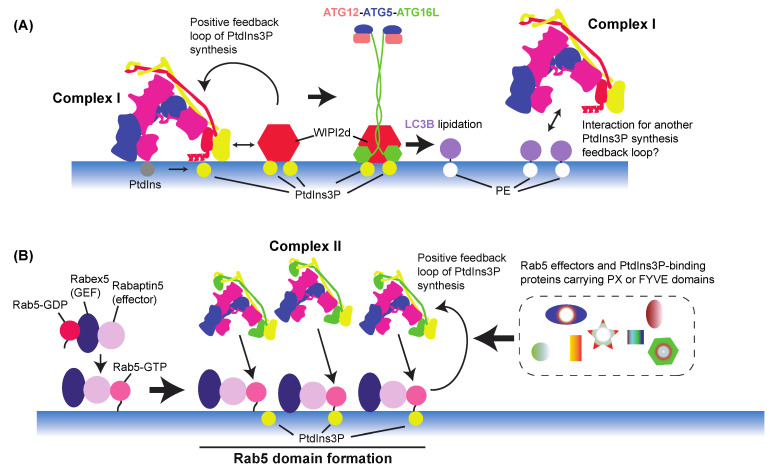
Positive PtdIns(3)P feedback mechanisms via complexes I (A) and II (B). (**A**) Complex I is activated by its effector WIPI2d, which leads to a positive feedback loop of PtdIns(3)P synthesis. WIPI2d also facilitates the LC3B lipidation by interacting with the ATG12–ATG5–ATG16L E3 complex. GABARAP and GABARAPL1, Atg8 family proteins, are known to preferably interact with complex I, but the consequence of this interaction remains to be seen. (**B**) On early endosomes, Rab5 interacts with its GEF-effector complex, the Rabex5–Rabaptin5 complex. This interaction switches the Rab5 nucleotide binding from GDP to GTP and its binding partner from Rabex5 to Rabaptin5. This interaction causes the formation of Rab5 domain (clustering), which is enhanced by PtdIns(3)P, indicating positive feedback between Rab5–Rabex5–Rabaptin5 and complex II. These Rab5- and PtdIns(3)P-enriched clusters recruit more Rab5 effectors and PtdIns(3)P-binding proteins carrying PX and FYVE domains.

**Table 1 cells-10-03124-t001:** Summary of effects of lipid physicochemical parameters on the activities and membrane binding of autophagy-related proteins. The lipid physicochemical parameters of ATG9/Atg9 vesicles are also involved.

Protein (Complex)	Packing	Curvature	Electrostatics	Reference
Atg1 complex binding to membranes	Not known	Binds more strongly to smaller vesicles	PS concentration did not affect membrane binding of Atg1 and Atg13, PtdIns(3)P increased Atg1 binding to membranes	[97,187]
ATG9/Atg9 vesicle property	High unsaturated lipids (yeast)	High curvature (30–60 nm)	<10% PS (yeast)	[81,196,197]
Human VPS34 complexes I and II	More active on GUVs composed of unsaturated lipids	More active on smaller vesicles, BATS domain binds to smaller vesicles more strongly	More active with higher PS; complex II interacts with high-PS-containing membranes more strongly	[24,26,45]
ATG2A/B/Atg2 membrane binding (MB), tethering (MT), and lipid transfer (LT)	Atg2 recognizes packing defects and binds to smaller vesicles. LT and MT by ATG2A–WIPI4 with DO (18:1–18:1) lipids are more efficient than PO (16:0–18:1) lipids	Stronger MB, MT, and LT on smaller vesicles (ATG2A/B/Atg2)	PS and PtdIns(3)P increase MB, MT, and LT of ATG2A and ATG2B, whereas they decrease MT and LT activity of Atg2	[198,199,200,201,202]
LC3B/Atg8 lipidation	More efficient on GUVs with DO (18:1–18:1) lipids than those with PO (16:0–18:1) lipids (LC3B)	Possible with Atg12–Atg5 on SUVs, requires Atg16–Atg12–Atg5 on GUVs (Atg8)	Not known	[203,204]
Membrane tethering and fusion by GATE-16 and GABARAP	Not known	Smaller vesicles, CL and DAG facilitate fusion	Not known	[205]
Tethering comparison between LC3B and GATE-16	Not known	LC3B > GATE-16 on 50 nm vesicles, GATE-16 > LC3B on 200–400 nm vesicles	Not known	[206]

## Data Availability

The data presented in this study are available on request from the corresponding author.

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
