# Peer review of "Activation Mechanisms of the VPS34 Complexes"

_cells, 2021, doi:10.3390/cells10113124_

Round 1

Reviewer 1 Report

This review on the activation mechanisms of the VPS34 kinase complex gives a comprehensive overview on the role of the PI3 kinase VPS34 and the associated proteins in both autophagy and endocytic trafficking.

It provides a broad overview of the biology of the VPS kinase complex, the lipid PtdIns3P, the effector proteins of PtdIns3P and their role in autophagy and membrane trafficking.

The review is very well written and covers the current state of the art in depth. It addresses both the original discoveries and the newest results from the literature. So far, it is probably one of the most exhaustive reviews of VPS34 and its regulators published to date, and will be an important reference for both experts in the field and a wider audience interested in phosphoinositides, autophagy and membrane dynamics.

However, one of the main concern that I have for this review is that it potentially covers too much ground.

The innocuous title implies that this review focusses on VPS34 and its regulation, but while reading it, the reader is presented with a cornucopia of information, which does not only focus on VPS34, but also covers general membrane biology, the influence of lipid charges and membrane packing on the properties of membranes, the role of membranes in health and disease, as well as a short overview of mTORC signalling and many other related topics.

This can potentially distract from the main topic – the regulation of the VPS34 kinase complex, since readers might expect an overview focussed on this topic, and the overall message might be hard to extract from the wealth of information.

However, this depth of information can also be a boon, as a reader will get a thorough introduction into most of the processes regulating autophagy and membrane trafficking. Thus, the review and the – exhaustively - referenced literature will be perfect to guide a novice in membrane biology to a thorough understanding of the biology and biophysics regulating membrane dynamics in autophagy, and will serve as a reference for the experts in the field, as the major findings of the last decades are collated and reviewed in one review.

Personally, this reviewer appreciates the depth of this review and the number of topics covered and would like to see it published as is, as it would provide a valuable resource for the experts in the field.. However, it is a challenging read due to the wealth of information, and  it will be up to the editors to decide if this review is focussed enough for the target group or if a shorter, more concise version would be better suited.

Apart from this, I have only minor comments:

  • The author should check the spelling of some gene names, e.g. line 266 “DFCP” should be “DFCP1”
  • Line 870 – “By using this character” – should this be “By using this characteristics”?

Taken togeher, I highly recommend publication, and I am looking forward to read the published version and give it to my students as an introduction to the field.

Author Response

First of all, I really thank this reviewer for reading this relatively long manuscript, and for finding values in it. As this reviewer understands, I thought that it would be beneficial for the readers to be introduced to general membrane biology and to some proteins/protein domains in addition to VPS34/Vps34 complexes. This has made the manuscript long, but my hope is that this manuscript would serve as an all-in-one review on VPS34/Vps34 activation mechanisms so that the readers do not need to look for other papers somewhere else. This has been consulted by the editor, and she has agreed not to shorten it. Also, the reviewer’s personal preference for keeping the current format is much appreciated and encouraging. The below are my point-to-point comments:

    The author should check the spelling of some gene names, e.g. line 266 “DFCP” should be “DFCP1”

- I appreciate the point the reviewer raised. I have corrected the gene/protein names that I have found to be corrected.

    Line 870 – “By using this character” – should this be “By using this characteristics”?

-This is absolutely true. I have corrected it. I appreciate this comment.

Best wishes,
Yohei Ohashi

Reviewer 2 Report

This is an exceptionally well-written manuscript. It comprehensively covers membrane biology of autophagy (including activation mechanisms of the VPS34 complexes). I truly enjoyed reading this review article; only have two minor comments before publication.

  1. The description ‘Also the expansion occurs via tubulation, which might be generated by the yeast Atg20-Atg24 complex’ in the figure legend for Fig 6 is a bit confusing. The figure indicates tubulation of phagophore edge but this is hard to imagine. Do you mean the formation of isolation membrane-associated tubules (IMATs)? If not, would ‘membrane remodeling (e.g. curvature stabilization to support phagophore expansion)’ be more appropriate to translate the in vitro data to in vivo situation?
  2. The following sentence ‘The phagophore is tethered and fused to become a closed autophagosome, which is facilitated by the Atg8 family proteins’ in the figure legend for Fig 6 may be misleading as autophagosomes closure is a membrane scission event (to generate two separate lipid bilayers from a single membrane structure) but not a fusion event. Given that ESCRT inhibition impairs autophagosomes closure despite the accumulation of LC3/ATG8 on the membranes (PMIDs: 30030437; 31010855; 31366282; 31519728), the ATG8 family proteins may not directly regulate the event.

Author Response

It is my great pleasure that this manuscript has been evaluated by this reviewer as “This is an exceptionally well-written manuscript.”. I thank this reviewer for reading this manuscript carefully, and for making appropriate comments. The below are my point-to-point comments:

1.   The description ‘Also the expansion occurs via tubulation, which might be generated by the yeast Atg20-Atg24 complex’ in the figure legend for Fig 6 is a bit confusing. The figure indicates tubulation of phagophore edge but this is hard to imagine. Do you mean the formation of isolation membrane-associated tubules (IMATs)? If not, would ‘membrane remodeling (e.g. curvature stabilization to support phagophore expansion)’ be more appropriate to translate the in vitro data to in vivo situation?

- I appreciate this appropriate comment. Regarding IMATs, they appear to occur at the ER, whereas the Atg20-Atg24 complex is mainly found in endocytic pathways. Therefore this time I have decided not to describe IMATs, although this possibility cannot be excluded. Whereas the second suggestion “curvature stabilization to support phagophore expansion” is a more likely possibility as previously described in PMID: 29114050. Because the cytoplasmic side of the yeast autophagosome does not contain much PtdIns(3)P, the most likely regions, where the Atg20-Atg24 complex targets, could be the edges of expanding phagophores, rather than the outer membrane regions, where Atg8-PE generates curvatures.I have accordingly revised the figure legend in Figure6. 

2.    The following sentence ‘The phagophore is tethered and fused to become a closed autophagosome, which is facilitated by the Atg8 family proteins’ in the figure legend for Fig 6 may be misleading as autophagosomes closure is a membrane scission event (to generate two separate lipid bilayers from a single membrane structure) but not a fusion event. Given that ESCRT inhibition impairs autophagosomes closure despite the accumulation of LC3/ATG8 on the membranes (PMIDs: 30030437; 31010855; 31366282; 31519728), the ATG8 family proteins may not directly regulate the event.

- I really appreciate this very important comment. I have revised the figure legend for Figure 6 to “During the expansion phagophores are tethered and fused, which is facilitated by Atg8 family proteins.”. Also Figure 6 has been replaced with a revised one.

Best wishes,
Yohei Ohashi